# SIRT1 accelerates the progression of activity-based anorexia

Timothy M. Robinette [1,2], Justin W. Nicholatos[2], Adam B. Francisco [2], Kayla E. Brooks[2], Rachel Y. Diao[2], Sandro Sorbi[3,4], Valdo Ricca[3,5], Benedetta Nacmias [3], Miguel A. Brieño-Enríquez[1] & Sergiy Libert [2,6✉]

Food consumption is fundamental for life, and eating disorders often result in devastating or life-threatening conditions. Anorexia nervosa (AN) is characterized by a persistent restriction of energy intake, leading to lowered body weight, constant fear of gaining weight, and psychological disturbances of body perception. Herein, we demonstrate that SIRT1 inhibition, both genetically and pharmacologically, delays the onset and progression of AN behaviors in activity-based anorexia (ABA) models, while SIRT1 activation accelerates ABA phenotypes. Mechanistically, we suggest that SIRT1 promotes progression of ABA, in part through its interaction with NRF1, leading to suppression of a NMDA receptor subunit Grin2A. Our results suggest that AN may arise from pathological positive feedback loops: voluntary food restriction activates SIRT1, promoting anxiety, hyperactivity, and addiction to starvation, exacerbating the dieting and exercising, thus further activating SIRT1. We propose SIRT1 inhibition can break this cycle and provide a potential therapy for individuals suffering from AN.

[1] Department of Obstetrics, Gynecology, and Reproductive Sciences, Magee-Womens Research Institute, University of Pittsburgh School of Medicine, Pittsburgh, PA 15213, USA. [2] Department of Biomedical Sciences, Cornell University, Ithaca, NY 14853, USA. [3] Department of Neuroscience, Psychology, Drug Research and Child Health, University of Florence, Florence 50139, Italy. [4] IRCCS Fondazione Don Carlo Gnocchi, Florence 50139, Italy. [5] Psychiatry Unit, Department of Health Sciences, University of Florence, Florence 50139, Italy. [6] Calico Life Sciences, South San Francisco, CA 94080, USA. ✉email: libert@calicolabs.com

Anorexia nervosa (AN) is characterized by a persistent restriction of food intake, leading to lowered body weight, constant fear of gaining weight, and psychological disturbances of body perception[1]. People suffering from AN are often diagnosed with additional ailments, such as anxiety, depression, and hyperactivity[2]. Certain activities and chronic stress are known to increase the risk of AN. For example, women who participate in sports that encourage lean shape and low body mass, such as gymnastics and figure skating, have been shown to be more likely to be diagnosed with AN[3]. Military servicewomen are also more likely to suffer from eating disorders compared to civilian women[4]. There is currently no pharmacological therapy for AN[1], despite having the highest mortality rate of any psychiatric disorder[2], which urges the search for therapeutics for treatment of this disease.

SIRT1 (silent mating-type information regulation 2 homolog 1) is a $NAD^+$ (nicotinamide adenine dinucleotide)-dependent enzyme, known to respond to stress and nutrient availability[5]. $NAD^+$ is an electron carrier, typically reduced to NADH when food is abundant. During prolonged fasting, starvation, or extreme dieting, $NAD^+$ levels rise due to an increase in both biosynthesis of $NAD^+$ and the oxidation of NADH to $NAD^+$[6]. This increase leads to the activation of SIRT1, which further represses the transcription of another $NAD^+$ consumer, PARP1, to increase the available $NAD^+$[6]. SIRT1 then translates these nutrient-limiting signals into changes in the activity of several transcription factors (TFs), which leads to changes in gene expression that guide cellular and organismal responses[7]. For example, calorie restriction induces foraging and hyperactivity in rodents, something that is not observed in animals that lack SIRT1[8], or have conditional knockout (KO) in the brain[9]. Conversely, animals that overexpress (OX) SIRT1 are more active in stressful situations[10]. These observations are especially intriguing, since hyperactivity, or excessive exercise, is universally observed in individuals suffering from AN. In addition to being linked to physical activity, SIRT1 is implicated in the development of addictions[11,12], anxiety[13,14], and depression[14–17], all of which are elements of AN, reinforcing the notion that this enzyme might play a key role in the pathology of AN. Furthermore, two recent studies showed an upregulation of SIRT1 expression in AN models of rodents[18,19], while another found an increase in SIRT1 protein levels in humans with AN[20], all of which further suggest a link between AN and SIRT1.

In this study, we demonstrate that the brain-specific KO of SIRT1 in mice is protective against the AN phenotypes when subjected to the activity-based anorexia (ABA) model. The brain-specific overexpression has the opposite effect and leads to worsened phenotypes. We confirmed our findings pharmacologically by inhibiting SIRT1 using selisistat and activating SIRT1 using SRT1720 and resveratrol. We suggest that SIRT1 inhibition results in the upregulation of the TF nuclear respiratory factor 1 (NRF1), which induces an upregulation of *Grin2a*, leading to sufficient NMDA (*N*-methyl-D-aspartate) receptor (NMDAR) function. We propose that SIRT1 inhibition with selisistat is a potential therapeutic target for AN.

## Results

**SIRT1 KO animals are protected from ABA.** AN is not naturally observed in mice; however, there are several commonly used models that recapitulate different aspects of AN. Two major models are the stress model and the ABA model[21]. Stress has a strong negative impact on eating behavior, leading to loss of weight and appetite. Mice subjected to the stress model show a loss of appetite due to hormonal imbalances, similar to what is seen in humans. The ABA model leads to multiple AN-like phenotypes and is induced by pairing running-wheel activity with restricted food access. When mice are subjected to ABA, they voluntarily induce self-starvation, rapidly lose body weight (Supplementary Fig. 1a, b), become hyperactive (Supplementary Fig. 1c), hypothermic, display loss of estrus, induction of hypothalamic–pituitary–adrenal axis signaling, distortion of circadian rhythm (Supplementary Fig. 1d), suffer from stomach ulceration, and will die if they are not removed from the model conditions. All of these phenotypes are similar to pathologies observed in humans suffering from AN.

To evaluate the impact of SIRT1 on AN-like behavior in mice, we subjected mice that either lack functional SIRT1 specifically in the brain (BSKO mice) or OX SIRT1 in the brain (BSOX)[14] to the ABA model. The BSKO mice express a catalytically dead SIRT1 protein, due to the deletion of exon 4[22], solely in the brain, using a *nestin-cre driver*[10] (Fig. 1a). BSOX animals OX SIRT1 specifically in the brain also using a *nestin-cre* driver[10], which causes constitutive overexpression of SIRT1 in both neurons and glia of these animals (Fig. 1a). BSKO mice are known to be smaller than their wild-type (WT) littermates[22] (Fig. 1b), but there is no difference between the baseline weight of BSOX and WT mice. We subjected the animals to ABA and found that the BSKO mice are protected, as they lose their weight slower than their WT counterparts (Fig. 2a). This protection is even more impressive

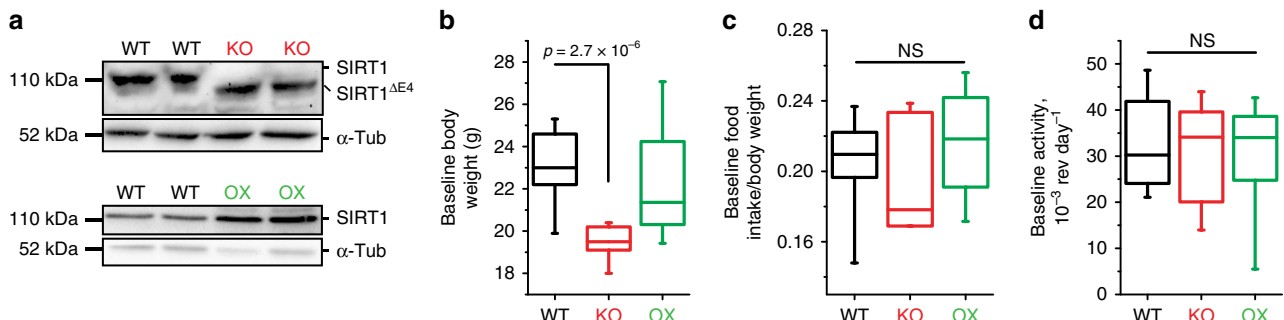

**Fig. 1 SIRT1 BSKO and BSOX baseline phenotypes. a** Brain-specific knockout (BSKO) mice lack functional SIRT1 in the brain, while the brain-specific overexpressing (BSOX) mice overexpress SIRT1 in the brain. This is representative of three independent experiments. **b** BSKO mice (red) are significantly smaller than their littermates, while there are no differences between WT (black) and BSOX (green) mice. **c**, **d** No differences were observed in baseline food intake or activity of WT, BSKO, or BSOX mice. *P* values were calculated using unpaired two-tailed *t* tests. The box-plots represent the median, 25th, and 75th percentiles of the data and the whiskers represent 5–95% of the data. Six mice were used in each cohort. NS, not significant. Source data are provided as a Source Data file.

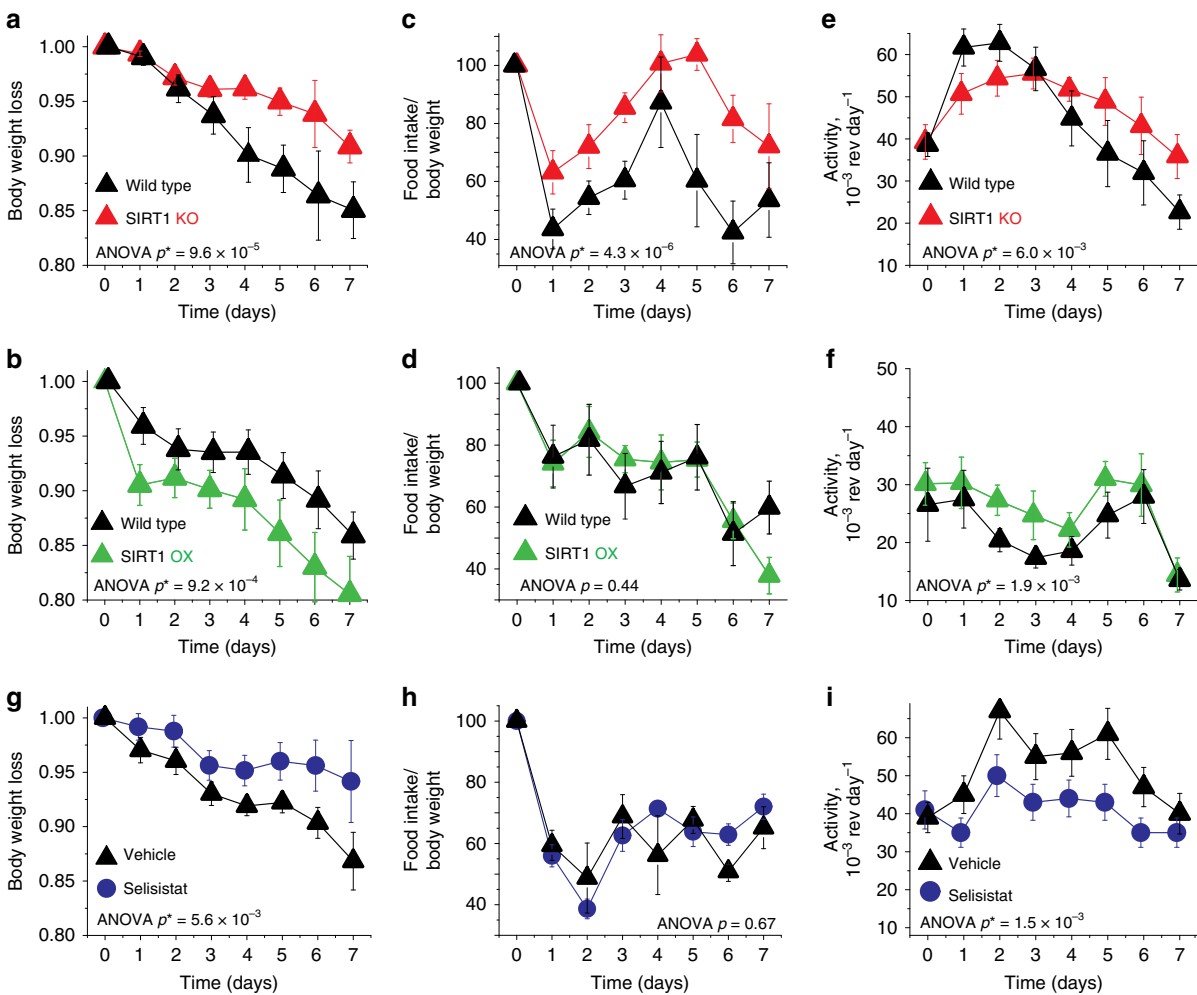

**Fig. 2 SIRT1 activity amplifies AN-like phenotypes in animals. a** BSKO mice (red) subjected to ABA are resistant to the model and lose weight slower compare to their WT littermates (black). Weight loss in fraction of initial body weight over time is presented. **b** BSOX mice (green) are more susceptible to ABA and showed faster weight loss. **c** The BSKO mice show protection from the decline in food intake typically observed during the ABA model. **d** BSOX mice show no difference in food intake compared to their WT littermates. **e** The BSKO mice also do not upregulate their activity as much as their WT littermates. **f** The BSOX mice have an elevated physical activity throughout the ABA model. **g** Selisistat-treated mice (blue), similarly to the BSKO mice, show protection from body weight loss throughout the ABA experiment. **h** Selisistat-treated mice consume roughly similar amounts of food during ABA. **i** Mice treated with selisistat do not upregulate physical activity during ABA. Statistical analysis was completed using two-way ANOVA with SEM error bars. $n = 6$ for the BSKO and BSOX mice and their WT littermates. $n = 6$ for the vehicle control and $n = 7$ for the selisistat-treated mice. Source data are provided as a Source Data file.

due to a recent report revealing that mice with lower baseline weight are more susceptible to ABA[23]. Conversely, BSOX animals were more susceptible to ABA and lost weight faster than their WT littermates (Fig. 2b). From a bioenergetics point of view, two major dysfunctions of AN are insufficient nutrient intake and elevated physical activity. At baseline, the amount of food consumed normalized to the animals' body weight, as well as the baseline activity, is identical between BSKO, BSOX, and their respective WT littermates (Fig. 1c, d). When subjected to the ABA model, we find that BSKO mice are protected from both dysregulations. These transgenic mice continue to eat significantly more food than their counterparts (Fig. 2c), suggesting a more robust feeding behavior control. We also find that BSKO mice are partially protected from an increase in their physical activity during ABA (Fig. 2e), suggesting that they might be resistant to exercise addiction caused by ABA. In this case the animals had a sharp decrease in activity a few days into the experiment due to physical exhaustion, wasting, and other pathologies associated with ABA[24]. The BSOX mice do not

exhibit a difference in food consumption compared to WT mice (Fig. 2d). However, they do show an increase in physical activity (Fig. 2f). These data suggest that SIRT1 controls several aspects of ABA pathologies.

**Pharmacological inhibition of SIRT1 is therapeutic against ABA.** In our previous experiments, SIRT1 dosage was altered in the brains of BSKO and BSOX mice (Fig. 1a) beginning from the embryonic day 7.5. It is possible that developmental effects or body weight differences had an impact in their susceptibility to ABA[23]. Therefore, to test if acute modulation of SIRT1 is sufficient to influence ABA phenotypes, we activated SIRT1 with SRT1720 and resveratrol and inhibited SIRT1 with selisistat (EX527). Resveratrol and SRT1720 have been shown to activate SIRT1 in different contexts, however, previous reports support SRT1720 as being a much more potent activator of SIRT1[25,26]. Mice treated with SRT1720 were indeed more susceptible to the ABA model than their WT counterparts (Supplementary

Fig. 2a–c). The mice treated with resveratrol did not show an increased susceptibility to the ABA model overall (Supplementary Fig. 2d–f); however they did tend to have a slightly lower body weight and food intake along with a statistically significant increase in activity levels. Selisistat is a specific SIRT1 inhibitor with partial blood–brain barrier permeability developed by Elixir, suitable for in vivo delivery, which was recently approved for use in human clinical trials[27,28]. We found that WT mice treated with selisistat are less susceptible to ABA than control animals (Fig. 2g). Unlike the BSKO mice, they had no change in food intake (Fig. 2h); however, they were protected from the ABA-induced hyperactivity, which likely aided in their weight retention (Fig. 2i). Taken together, these data demonstrate that inhibition of SIRT1 (genetic or pharmacological) is sufficient to ameliorate AN-like pathology and slow down the associated weight loss.

**NR2a is associated with hyperactivity and AN.** We have previously demonstrated that SIRT1 induces Mao-A expression, by deacetylating and thus activating its transcriptional activator—NHLH2[14], which inadvertently results in diminished serotonin levels, and anxiety-like behavior. Interestingly, higher expression of Mao-A gene variants[29] and reduced levels of serotonin[30] are also associated with anxiety and AN in humans. In this way, the SIRT1–Mao-A axis could be involved in the mechanism by which SIRT1 inhibition protects against ABA. However, as we demonstrated using the ABA animal model (Fig. 2), SIRT1 inhibition also protects against AN-associated hyperactivity and decrease in food intake, which suggests the existence of additional mechanisms.

To identify additional players, we used a disease–gene association database to identify top genes associated with hyperactivity and/or feeding behavior. We cross-referenced the resulting list with genes reported to be involved in psychiatric disorders[31]. We identified *Agrp*, *Drd1*, *Drd2*, *Gria1*, *Grin2a*, and *Pomc* as top candidates and investigated their expression in the BSKO and BSOX mice. The only gene with altered transcription levels in BSKO and selisistat-treated mice, with the opposite expression in BSOX mice, was *Grin2A* (glutamate ionotropic receptor NMDA-type subunit 2A; Fig. 3a, Supplementary Fig. 3a). Grin2A encodes the NR2a subunit of NMDARs, which are associated with hyperactivity disorders[32,33] and eating disorders, including AN. NMDARs are pivotal in determining behavior, and among many other functions, control addiction formation and mood[34]. Interestingly, anti-NMDA receptor encephalitis, a condition where the immune system attacks and destroys its own NMDARs, has recently been shown to result in a classical AN-like syndrome in humans[35]. Functional NMDAR complexes are composed of four subunits, two from Grin1 and two from Grin2, as well as other adaptor, signaling, and cell-adhesion proteins[36]. Grin2 encodes regulatory subunits, two major forms of which are NR2A and NR2B. They have different biochemical properties and thus alter receptor ligand sensitivity. NMDAR subunit composition is also known to be altered by calorie restriction[36]. These associations and the altered transcription levels of *Grin2a* suggest that SIRT1 may be involved in controlling NMDAR subunit composition.

**Grin2a is suppressed by SIRT1.** We tested the expression levels of Grin1 and Grin2 subunits in the brains of WT, BSKO, and BSOX animals. *Grin2A* was still the only gene that was significantly altered when knocking out or pharmacologically ablating SIRT1, with the opposite effect when overexpressing SIRT1 (Fig. 3a, b; Supplementary Fig. 3b). Consistent with this, we found elevated levels of Grin2A protein in brains of BSKO mice (Supplementary Fig. 3c). To test if SIRT1 influences Grin2A

transcription directly, we engineered a reporter construct that contained 2 kb of the Grin2A promoter in front of the luciferase gene. This reporter construct was transfected into the neuronal cell line, Neuro-2a, with constructs that OX or knockdown (KD) the expression of SIRT1. We found that when SIRT1 is over-expressed in Neuro-2a cells (Fig. 3c), the activity of Grin2A promoter decreases (Fig. 3d). Conversely, when we reduce SIRT1 abundance in Neuro-2a cells using short hairpin RNA (Fig. 3c), the activity of the Grin2A promoter increases (Fig. 3d). These data suggest that SIRT1 directly influences abundance of the Grin2A subunit of NMDAR, and thus, likely its function.

**NRF1 mediates the impact of SIRT1 on Grin2A transcription.** SIRT1 does not have any DNA-binding motifs, and often controls expression by interacting with TFs. Therefore, we used a combination of prediction algorithms[37,38] and ChIP-seq databases[39,40] to seek TFs that bind to the Grin2A promoter. The TFs of interest were those that are known to be both expressed in the brain and involved in metabolic processes. Using this approach, we narrowed the search to eight transcription factors: EGR1, HES1, NFATc3, NFκB, NRF1, RxRβ, TFB2M, and USF1. Using small interfering RNAs (siRNAs) for each individual TF, we transfected Neuro-2a cells along with the 2 kb Grin2A promoter and the T1OX or T1KD constructs. The only TF that eliminated the impact of SIRT1 on Grin2A transcription when knocked down was NRF1 (Fig. 3e). NRF1 is known to regulate neurite outgrowth, and similarly to Grin2A, has been associated with hyperactivity[41] and AN[42]. It has also been shown to be induced by PGC-1α (PPARG coactivator 1 alpha), which is upregulated by SIRT1 activity[43]. Interestingly, a SIRT1–NRF1 interaction has already been documented in the liver[44]. To confirm that NRF1 is bona fide mediator of SIRT1 action on Grin2A, we mutated the binding site for NRF1 on the 2 kb Grin2A-luciferase reporter. The mutation completely abrogated the responsiveness of the construct to SIRT1 dosage (Fig. 3f). These data suggest that SIRT1 controls expression of Grin2A via NRF1.

**Grin2A partially mediates the impact of SIRT1 on ABA.** To confirm that Grin2A is involved in the SIRT1-AN pathway, we generated Grin2A KO (G2A KO) mice using CRISPR/Cas9 technology (Fig. 4a–c). We induced a deletion of 13 nucleotides within the first exon of Grin2A, which resulted in a shift of the reading frame. The mutated messenger RNA (mRNA) is still expressed at nearly WT levels (Fig. 4b); however, no protein is being produced (Fig. 4c). The G2A KO mice show no difference in baseline body weight or their food intake compared to their WT littermates (Supplementary Fig. 4a). However, the G2A KO mice do show an increase in their baseline activity (Supplementary Fig. 4a), consistent with Grin2a's previous associations with hyperactivity. We subjected the G2A KO mice and their WT littermates to the ABA model and observed that the G2A KO mice lost significantly more weight than the WT mice (Supplementary Fig. 4b). The two cohorts did not show a difference in the food intake (Supplementary Fig. 4c); therefore, their decrease in body weight is likely due to the increase in activity seen throughout the experiment (Supplementary Fig. 4d). To determine if the inhibition of SIRT1 would be nullified in the G2A KO mice, we subjected two cohorts of G2A KO mice to ABA. The first cohort was treated with selisistat, while the other was given a vehicle control. Unlike in WT mice, where selisistat significantly slowed the progression of weight loss during ABA (Fig. 2g), G2A KO mice had little or no benefit from selisistat treatment (Fig. 4d–f). Selisistat-treated G2A KO mice had similar weight loss dynamics (Fig. 4d), food consumption (Fig. 4e), and only marginally reduced activity (Fig. 4f). These data further suggest

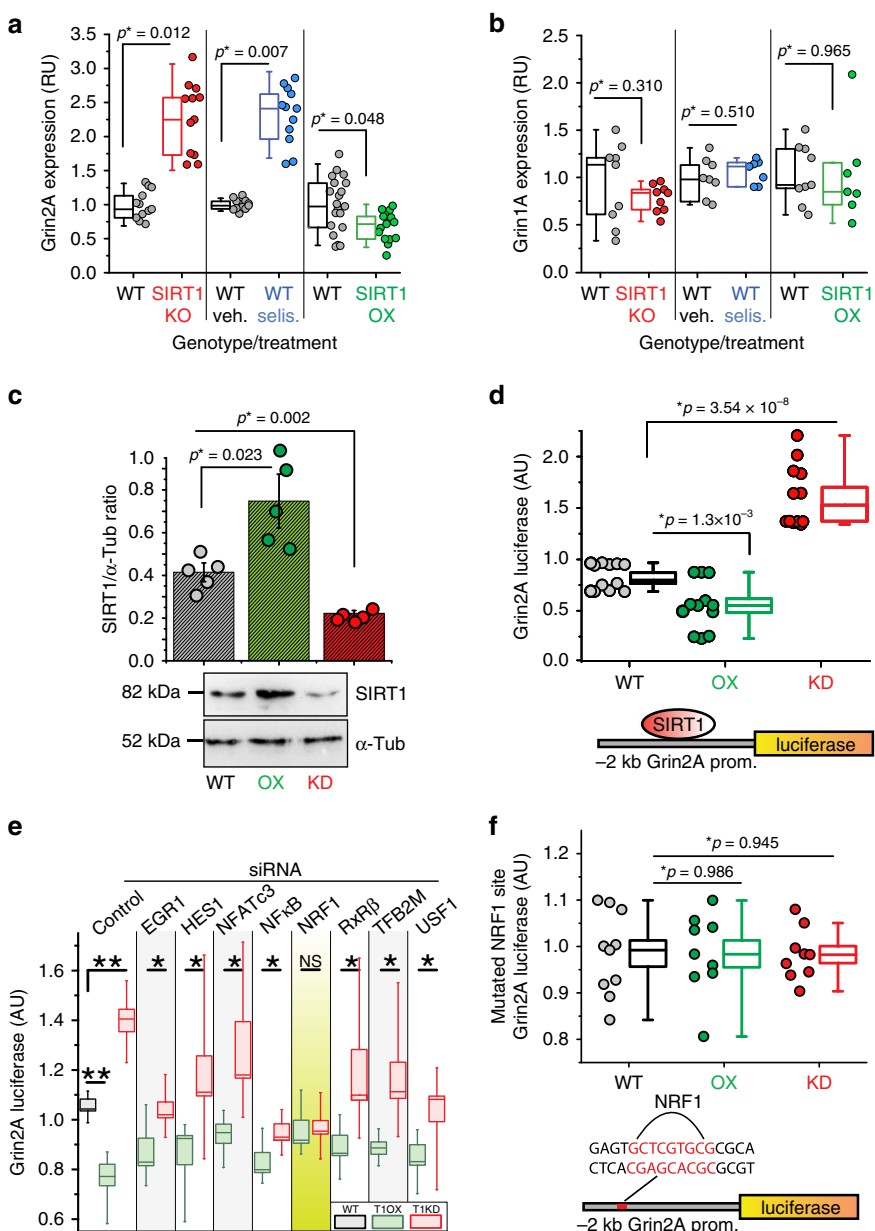

**Fig. 3 SIRT1 activity, mediated by NRF1, suppresses the expression of Grin2A. a** RT-qPCR expression analysis of Grin2A in the cortex of BSKO- (red), BSOX- (green), and selisistat- (selis.) (blue) treated mice, $n \geq 12$. **b** Expression analysis of *Grin1A* gene in the same mice as in **a**, $n \geq 7$. **c**, **d** Modulation of SIRT1 abundance in vitro in Neuro-2a cells directly influences the activity of the Grin2A promoter. **c** SIRT1 protein levels in Neuro-2a cells are shown, together with an example of a typical SDS-PAGE, $n \geq 5$. **d** The activity of the Grin2A-luciferase reporter in the same cells as in **c** is presented, $n \geq 12$. **e** Using siRNAs for eight transcription factors (TFs), we transfected Neuro-2a cells along with the 2 kb Grin2A promoter, and T1OX or T1KO constructs. NRF1 was the only TF that eliminated the impact of SIRT1 on Grin2A transcription, $n \geq 6$. **f** Transfecting Neuro-2a cells with a Grin2A promoter with its NRF1-binding site mutated and the T1KO and T1OX constructs resulted in a complete loss of responsiveness to SIRT1, $n \geq 9$. P values were calculated using unpaired two-tailed *t* tests, *$P < 0.05$, **$P < 0.005$. Error bars are SEM. The box-plots represent the median, 25th, and 75th percentiles of the data and the whiskers represent 5–95% of the data. NS, not significant. Source data are provided as a Source Data file.

that Grin2A partially mediates the impact of SIRT1 on ABA progression and AN-like pathologies.

**SNPs in SIRT1 are associated with AN susceptibility.** To investigate if SIRT1 gene variants are associated with the risk of AN in humans, we analyzed the SIRT1 region in 1001 patients, who were confirmed to suffer from AN (NIH dbGaP dataset phs000679.v1.p1)[45] and compared it to the same region among 1987 control subjects[46], who were matched by gender and controlled for age and ancestry (Supplementary Fig. 5). SIRT1 gene

variants marked by common single-nucleotide polymorphisms (SNPs) rs730821 and rs10997881 have statistically significant association with increased risk of AN, p values of $2.3 \times 10^{-5}$ and $1.1 \times 10^{-4}$, respectively. These associations were statistically significant after adjusting for multiple testing using Bonferroni correction and permutation analysis (Supplementary Fig. 5a, c). Noteworthy, we found that four SNPs in the C terminus of SIRT1 were in a strong linkage disequilibrium (LD) (Supplementary Fig. 5b), forming a block that has a significant association with the susceptibility to AN. The C-allele of rs2273773, which we found to associate with AN (Supplementary Fig. 5c), has

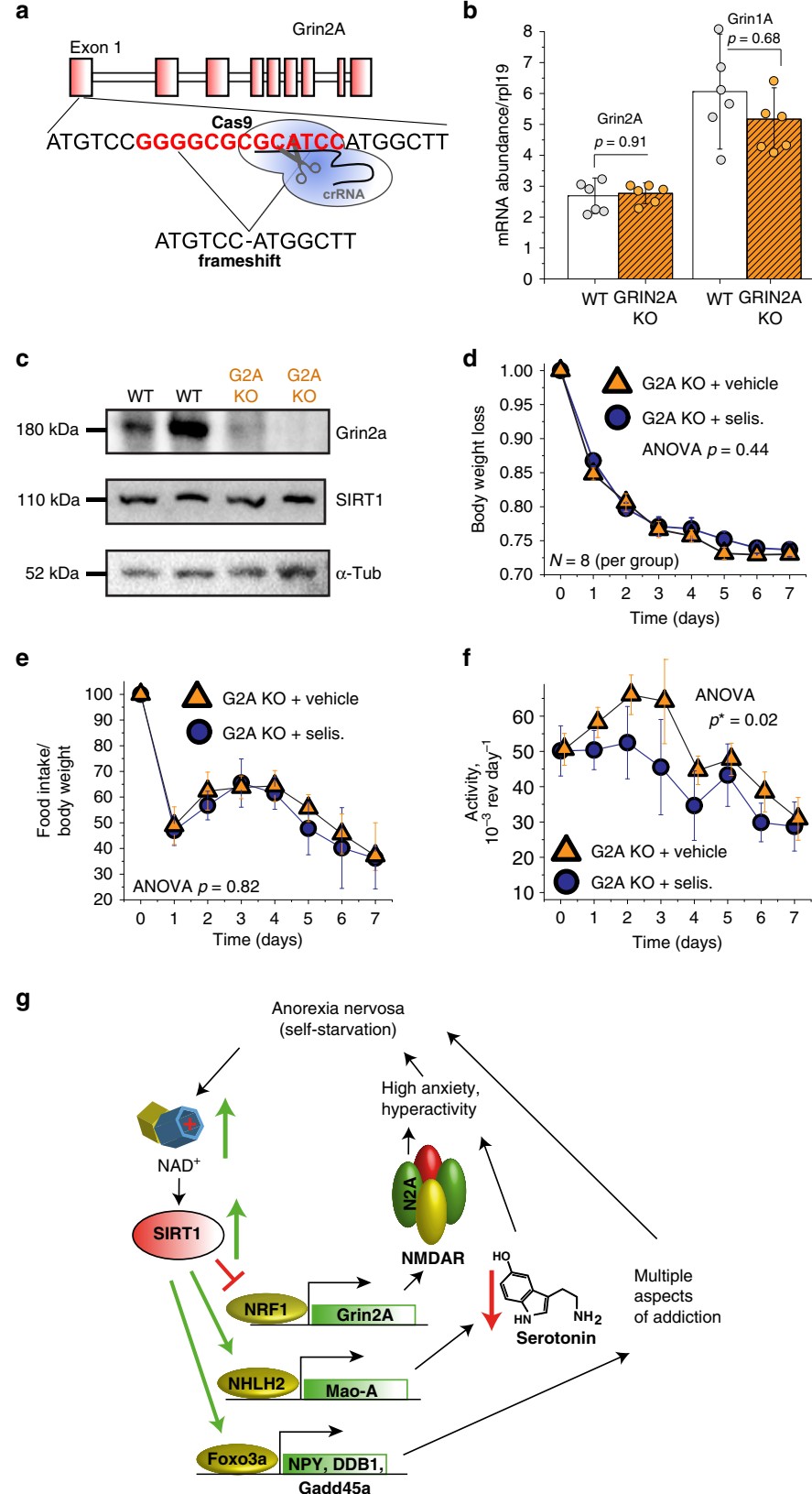

previously been associated with protection from seasonal variation in weight among 8028 adults from Finland, and elevated body mass index in a Dutch population[47]. Additionally, it was shown that lower expression of SIRT1 is associated with heavier weight, which is protective against AN[48]. When verifying the

SIRT1-AN association in an independent cohort of 112 anorexic subjects of European descent[49] compared with 3937 control subjects[50,51], rs730821 was again found to be associated with AN (Supplementary Fig. 5c), further supporting the notion that SIRT1 plays a role in controlling weight homeostasis and AN

**Fig. 4 Grin2A partially mediates multipronged action of SIRT1 in progression of ABA. a** Schematic representation of Grin2A knockout mouse generation using CRISPR/Cas9 technology. **b** RT-PCR quantification of Grin2A (orange) mRNA in brains of transgenic mice is presented. Truncated Grin2A mRNA is expressed at nearly WT levels. P values were calculated using unpaired two-tailed t tests, error bars are SEM, $n = 6$. **c** SDS-PAGE analysis of Grin2A expression in brains of Grin2A knockout mice. No Grin2A protein is expressed in transgenic line of mice. This is representative of three independent experiments. **d** Grin2A knockout mice (orange) do not benefit from pharmaceutical SIRT1 inhibition during ABA progression. Both selisistat- and saline-treated animals lose weight at identical rate (contrast to WT mice (blue) treated with selisistat, shown in Fig. 2g, and **e** have similar food consumption. **f** Selisistat injections have modest impact on ABA-induced hyperactivity in Grin2A-null background. Statistical analysis is done using two-way ANOVA with SEM error bars, $n = 8$. **g** Proposed multipronged mechanism of SIRT1 action in progression of AN. Source data are provided as a Source Data file.

pathogenesis. The association with rs10997881 was not able to be replicated; however, it could be interesting to test again in future studies with a higher number of samples, especially given the possibility of an LD block.

Individuals suffering from AN do not recognize that their starvation and excessive exercise is harmful; therefore, it is difficult to diagnose and gather data from them[1]. There have been five AN GWASs completed, however, due to the low sample size, three of them were underpowered and found no significant loci[45,52,53]. The fourth GWAS found the first genome-wide significant locus on chromosome 12[54]. The fifth GWAS was unable to replicate the locus found from the previous study, but they did find eight other loci and confirmed our implication of the combination of metabolic and psychiatric origins of AN[55]. Unfortunately, more samples will need to be collected to replicate and confirm these loci and to identify more correlated loci. As awareness and diagnosis of AN continues to improve, so will the ability to gather enough data to find more genes involved in AN using the unbiased GWAS approach.

## Discussion

We previously demonstrated how linking behavior to food availability is beneficial for species survival[14], as well as the role SIRT1 plays in this link. However, in humans, where calorie restriction can be voluntary, due to social or career-related reasons, this link might result in pathological behavior, such as AN. Upon dieting, individuals predisposed to AN, will thrive on the feeling of being much more active and attentive[56]. This will result in a high-like state, which allows them to continue their daily lives and consider their behavior beneficial. Unfortunately, they become accustomed to this state of hyperactivity and anxiety, paralleling features of addictive disorders[57], and will need to starve themselves progressively more, leading to the deterioration of their bodies.

We suggest that the initial decision to diet or increase exercise intensity leads to elevated levels of NAD$^+$, which in turn activates SIRT1. Sustained, hyper-activated SIRT1 results in compulsive hyperactivity, need to over-exercise, and anxiety, all of which lead to further self-starvation, and increased activation of SIRT1. This positive feedback loop could be one of the vicious cycles that manifest as AN (Fig. 4g). We propose a mechanism, where AN-related hyperactivity results, in part, from SIRT1 interacting with the transcription factor NRF1. This interaction results in the suppression of Grin2A, which likely disrupts NMDAR function. Previous work completed by us and other groups suggests that modulating SIRT1 could lead to anorectic behaviors via additional pathways. For example, we have previously shown that SIRT1 activates Mao-A, which leads to lower serotonin levels and heightened anxiety[14]. Work completed by other groups also suggest that the need to exercise could be due to SIRT1 activating Foxo3A, which has been shown to accelerate the establishment of addictive behaviors[11,12]. Interestingly, NAD$^+$ levels and SIRT1 activity decline with age[58], making SIRT1 hyper-activation less likely in older individuals. This is consistent with epidemiological data showing the chance of developing AN decreases with age.

Therefore, attempting to OX SIRT1 to extend youthfulness of the brain could also elevate the risk of AN.

SIRT1 is known to modulate the endocrine system via the hypothalamic-pituitary axis (HPA) during physiological adaptation to diet[59]. When SIRT1 is deleted either in outbred mice or as a heterozygous deletion, the mice are unable to properly adapt to dietary challenges such as calorie restriction or high fat diet[60]. It is interesting that in ABA this failure to adapt seems to lead to a positive outcome for the BSKO mice by preventing a decrease in food intake or increase in activity. Modulating SIRT1 has also been shown to have effects on the weight of mice. As mentioned above, BSKO mice are known to weigh less than their WT littermates[22]. Additionally, deletion of SIRT1 specifically in hypothalamic agouti-related peptide (AgRP) neurons results in mice with a lean body mass[61]. In contrast, overexpression of SIRT1 in pro-opiomelanocortin (POMC) and AgRP neurons in the arcuate nucleus of the hypothalamus has been shown to protect against age-associated weight gain[62]. Further insight into how SIRT1 affects the mice during ABA could be gained by studying changes in the HPA. Additionally, lowered activity in multiple regions of the human brain have been implicated in the progression of AN, including the cingulate cortex[63], left hypothalamus, antero-ventral striatum[64], and anterior insula[65]. Changes in the activity of nucleus accumbens, amygdala, substantia nigra, and cerebellum have also been reported to precede ABA in animal models[66]. Modulating SIRT1 specifically in these areas of the brains as well as specific cell types could be beneficial to further investigate changes in both cell activity and the local microenvironment during ABA.

The consequences of SIRT1 activation that we describe here, and those established by others[7–9,11,12,14,16,48,67,68], overlap closely with physiological changes associated with AN pathologies, which supports our hypothesis of AN development (Fig. 4g). We have provided evidence, via the ABA model, that SIRT1 upregulation accelerates the development of anorectic behaviors. While multiple environmental and genetic factors likely contribute to AN, our results indicate that we may be able to modulate AN phenotypes by downregulating SIRT1. However, further studies on will need to be completed to fully implicate SIRT1 in AN in humans.

## Methods

All resources generated during this study will be or already are openly shared with the research community.

**Animal experiments and transgenic mice**. All procedures were performed according to guidelines and under supervision of the Institutional Animal Care and Use Committee (IACUC) of Cornell University. All experiments were authorized by IACUC prior to their initiation. For all the tests, we used 3-month-old C57BL/6J male and female mice. All transgenic animals were compared to their corresponding WT littermates. Based on variability of prior collected data, necessary sample sizes were estimated using power analysis and reported on figures and in the text. Assignment to treatment groups was done at random using animal's ear-tag numbers and completed separately for males and females. Prior to the behavioral experiments, all mice were handled daily for 2 weeks to eliminate the influence of stress and anxiety on the experimental outcomes.

During the ABA protocol, animals were single housed, and provided with low profile running wheels (Med Associates, Inc., St. Albans, VT). Each wheel was equipped with magnetic rotation counter, which were electronically monitored by a computer using PCI Input/output board (PCI-DIO-120, Access I/O Products, Inc., San Diego, CA). Before the ABA protocol, all animals were entrained with running wheels for 10 days on 12-h dark:12-h light regimes. Baseline weight, food consumption, and activity were measured from days 6 to 10 during entrainment. For the pharmacological modulation of SIRT1, we pretreated animals for 3 days with a daily dose of 10 mg kg-1 selisistat, SRT1720, or resveratrol via intraperitoneal injection before initiation of the ABA protocol. We continued the daily injections for the duration of the ABA protocol. During the ABA protocol, experimental animals had ad libitum food for 3 h (during light phase, from ZT 4 until ZT 7), after which the food was removed. Control animals had continuous access to food, with running wheels present, or identical feeding regime as ABA animals, but with running wheels blocked from rotating. Animals that lost more than 30% of their initial body weight were considered terminal and removed from the study. The ABA protocol was optimized for the WT mice to lose ~85% of their body weight during the experiment to prevent mice from dying or needing to be pulled from the experiment. Exclusion criteria had been pre-determined before the start of experiments; however, no animals were excluded from the ABA studies.

**Conditional SIRT1 mice**. To obtain BSKO animals, we crossed SIRT1 conditional deletion allele mice, which have exon 4 of SIRT1 flanked by loxP sites[10], to a nestin-cre carrying strain of mice (JAX Stock# 003771). Animals were genotyped by PCR during weaning using the following primers: F-5′-GGTTGACTTAGGTCTTGTC TG and R-5′-CGTCCCTTGTAATGTTTCCC. Conditional SIRT1-overexpressing mice (BSOX) contain a SIRT1-overexpressing construct, preceded by STOP in all six frames cassette, flanked by loxP sites[10]. To OX SIRT1 specifically in the brain, these transgenic mice were crossed to nestin-cre line of mice (JAX Stock# 003771). Animals were genotyped by PCR using the following primers: F-5′-GCA-CAGCATTGCGGACATGC, F-5′-CCCTCCATGTGTGACCAA GG, and R-5′-GCAGAAGCGCGGCCGTCTGG. To determine the presence of the Cre transgene, animals were genotyped by PCR using the following primers: F-5′-AAGAACCTGATGGACATGTT and R-5′-TTTAGTTACCCCCAGGCTAA GTGC. Animals positive for both transgenes had SIRT1 dosage modified in the brain, and animals positive for only one transgene or negative for both transgenes served as littermate controls.

**G2A KO mice**. To disrupt Grin2a, we used the MIT CRISPR Design tool to analyze exon 1 of its sequence. This tool selects the potential cut sites to minimize genome-wide off-target effects. The most optimal target identified was 5′-ATGTCCGGGG CGCGCATCCA. A single guide RNA (5′-AAUGUCCGGGGCGCGCAUCCAGU UUUAGAGCUAUGCU) was synthesized by Integrated DNA Technologies and injected with standard tracrRNA and recombinant Cas9 enzyme into C57B/6 zygotes. Resulting zygotes were incubated to morula stage and implanted into pseudopregnant female mice. Born pups were screened for truncated Grin2A sequence by PCR and sequencing. Founders positive for Grin2A truncation were backcrossed to WT C57B/6 animals and then to each other to generate homozygous Grin2A nulls. Animals were genotyped by PCR using the following primers: F-5′-GGAGCAGGCAACCGGCTTG and R-5′-AGATGGGGAT-GAAAAGTCTGTG. Genomic DNA from each animal used in the study was sequenced to confirm Grin2A disruption.

**SIRT1 SNP genotyping**. Genomic DNA from the 112 Italian anorexia cohort cases[49] was de novo genotyped using specific Taqman (Applied Biosystems) probes for SNPs rs730821 and rs10997881. Genetic variants were assigned using an end-point quantitative PCR, which was performed using a Roche LightCycler 480 system, according to the manufacturer's instructions. The initial study was approved by the human research ethics committee of Florence University School of Medicine. The other cohorts had been genotyped using chips[45,46,49–51].

**RNA expression/quantitative real-time PCR**. To quantify gene expression, total RNA was extracted from the cortex or cultured cells by homogenization in Trizol Reagent (Invitrogen). Complementary DNA was prepared using the SuperScript III Kit (Thermo Fisher Scientific) and poly-T primers, according to the manufacturer's guidelines. The abundance of specific mRNA in samples was quantified using the SYBR Green detection method on Bio-Rad CFX96 optical cycler system. Relative expression of specific genes was calculated using the Cp method, and ribosomal RPL19 RNA as normalizer. The following primers were used in this study:
AGRP: 5′-AGAGTTCCCAGGTCTAAGTCTG and 5′-GCGGTTCTGTGGAT CTAGCA; Drd1: 5′-CACGGCATCCATCCTTAACCT and 5′-TGCCTTCGGAG TCATCTTCCT; Drd2: 5′-ACCTGTCCTGGTACGATGATG and 5′-GCATGGCA TAGTAGTTGTAGTGG; GRIA1: 5′-AAAGGAGTGTACGCCCATCTTTG and 5′-TGTCAACGGGAAAACTTGGAG; GRIN2A: 5′-ACGTGACAGAACGCGAAC TT and 5′-TCAGTGCGGTTCATCAATAACG; POMC: 5′-ATGCCGAGATTCTG CTACAGT and 5′-CCACACATCTATGGAGGTCTGAA; RPL19: 5′-ATGAGTAT GCTCAGGCTAACGA and 5′-GCATTGGCGATTTCATTGGTC; GRIN1A: 5′-A GAGCCCGACCCTAAAAAGAA and 5′-CCCTCCTCCCTCTCAATAGC; GRIN 1B: 5′-ATGCACCTGCTGACATTCG and 5′-TATTGGCCTGGTTTACTGCCT;

GRIN2B 5′-GCCATGAACGAGACTGACCC and 5′-GCTTCCTGGTCCGTGT CATC.

**Western blot analysis and protein detection**. Tissues or cells were lysed in TNT buffer (20 mM Tris, 150 mM NaCl, 1% Triton X-100) and supplemented with protease inhibitors (Roche, Cat# 4693116001). The mixture was centrifuged at $10,000 \times G$ for 5 min and the supernatant was taken. Protein levels were standardized using a Bradford protein assay. Protein was mixed with sodium dodecyl sulfate and electrophoresed in a 5% acrylamide gels. Proteins were transferred to a PVDF membrane (0.45 μM) and the membrane was then immunoblotted with antibodies, diluted at concentrations recommend by the manufacturer, against the specific proteins being examined. The following antibodies were used in the study: SIRT1 (1:1000) (Millipore 04-1557), α-tubulin (1:5000) (ABCam ab7291), and GluN2a (1:1000) (Cell Signaling 4205) antibodies. The images presented are representative of three independent experiments. Full scans of the western blots are available in Supplemental Fig. 6.

**GRIN2A promoter cloning**. Two kilobases of the Grin2a promoter (5′ from the ATG) was amplified with high-fidelity Phusion DNA polymerase from the genomic DNA of WT C57B/6 mice. The following primers were used: F-5′-GGTAC-CAGCTCCTGGTCGCACAA and R-5′-CTCGAGTAGGGTCCCTG-TAAGGTGGA. These primers also introduced KpnI and XhoI restriction sites at the 5′ and 3′ ends, respectively. The amplicon was digested with KpnI and XhoI and directionally cloned into a pGL3 basic luciferase reporter (Promega). pGL3:Grin2A clones were screened by double digest and verified by DNA sequencing. Empty basic pGL3 plasmids were used as negative controls, and the luciferase driven by cytomegalovirus (CMV) promoter was used as a positive control.

**Luciferase reporter activity assays**. Luciferase-mediated promoter activity studies were performed in Neuro-2a (ATTC® CCL-121™) cells, which were cultured in Dulbecco's modified Eagle's medium containing 10% fetal bovine serum, supplemented with Pen/Step and Plasmocin. From a large pool of cells, 20,000 cells were plated in 96-well tissue culture plates and were transiently transfected with appropriate plasmids 32 h later. In a typical experiment, 50 ng of pGL3:GRIN2A Firefly Luciferase reporter, 0.5 ng of Renilla Luciferase reporter (used as a transfection control), and 100 ng of CMV-SIRT1 (SIRT1-overexpressing construct), shSIRT1 (short hairpin-suppressing SIRT1 translation construct), or CMV-GFP (negative control) plasmid were added to each well. After 24 h, luciferase activity was measured using the Dual Luciferase Reporter assay (Promega) and a Bio-Tek plate reader according to the manufacturer's manual. All data shown are representative of three independent experiments.

**siRNA assays**. siRNA experiments were performed using TriFECTa® Dicer-Substrate RNAi (Integrated DNA Technologies). The exact experimental setup from the luciferase reporter activity assays were used with the following exception: In addition to pGL3:GRIN2A, Renilla, and the appropriate SIRT1 plasmid, 10 μM siRNA was added for EGR1, HES1, NFATc3, NFκB, NRF1, RxRβ, TFB2M, USF1, or EGFP (negative control) to each well.

**Directed mutagenesis**. The NRF1 site was mutated on the pGL3:GRIN2A promoter using the QuickChange Site-Directed Mutagenesis Kit. The primers used were forward: tgctgaggcggccgagatttttttttcgcagcacgccccattgc and reverse: gcaatggggcgtgctgcgaaaaaaaaaatctcggccgcctcagca. The mutated plasmid was sent for sequencing to confirm the correct mutation was made and to confirm its fidelity. The same parameters were used as in the luciferase reporter activity, where 100 ng of the mutated plasmid was transfected into Neuro-2a cells along with pGL3: GRIN2A and Renilla.

**Statistical analysis**. Unless otherwise noted, all the results are presented in the format of mean ± standard error of the mean. For pairwise comparisons, relevant to data analysis, two-tailed Student's t test was used and is usually reported on figures using asterisk for statistically significant p values, which are explicitly stated throughout. The t test was calculated assuming equal variance if variance of compared samples was similar. In cases where serval variables are influencing the same measured values, such as genotype and time influencing animal's activity, two-way analysis of variance (ANOVA) analysis was performed. In these cases, p values are reported in figure legends as ANOVA. In the event a mouse had to be pulled from an experiment or died, creating unbalanced data, a Type-III sums of squares test was performed. Where appropriate, further post hoc statistical tests were performed. Calculations were performed using licensed and registered copy of Microsoft Excel or the open-source free statistical software R, with the Bioconductor package.

For genotype association studies, seven SNPs were selected due to their prior association with metabolic disorders[69–74]. SIRT1 SNPs associated with metabolic disorders but not on the chips were excluded. A combination of R and p-link software was used to create a linear regression model. We required that samples had Hardy–Weinberg p values $> 10^{-7}$, <30% SNP missingness, <10% individual missingness, and >0.01 minor allele frequency. We also corrected for individual

age, sex, race, and the first two genotype principle components as covariates. Ancestry of origin was genetically inferred[45,46,49–51]. Bonferroni correction and 100,000 rounds of permutation analysis were used to account for multiple testing. To show the distribution of all the data, data are presented as box-plots with data points nearby, or bar graphs with data points overplayed on all the figures, where applicable.

**Gene–disease association studies**. The DisGeNET[31] database was used to analyze certain phenotype–gene association data and create testable hypothesis as described in the main text. The following criteria were used in the analyses (C-numbers are publicly available, database-specific identifiers): *Anorexia Nervosa* (C0003125), ADHD (OMIM# 143465, C1263846), hyperactive behavior (C0424295), obesity (C0028754), depression (C0011581). Gene expression co-regulation calculations were performed using NCBI publicly available datasets (GEO datasets) GSE39551 and GSE28790. Calculations were performed using a combination of licensed and registered copy of Microsoft Excel, Origin Pro, and a free, open-source statistical software R.

**Reporting summary**. Further information on research design is available in the Nature Research Reporting Summary linked to this article.

## Data availability

The source data underlying Figs. 1a–d, 2a–i, 3a–f, and 4a–f and Supplementary Figs. 1a–d, 2a–f, 3a–c, 4a–d are provided as a Source Data file. Data from dbGaP (https://www.ncbi.nlm.nih.gov/gap/) and Disgenet (https://www.disgenet.org/) were used in this study. All data supporting this study are included in this article and its supplementary information files, with raw data available from the corresponding author upon request.

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

## Acknowledgements

S.L., T.M.R., and J.W.N. were in part supported by a grant from American Federation for Aging Research (AFAR, grant # 2015-030). S.L. received seed grant funding from the Cornell University Center for Vertebrate Genomics. M.A.B.E was supported by the Eunice Kennedy Shriver National Institute Of Child Health & Human Development of the National Institutes of Health under the award K99HD090289 and R00HD090289. T.M.R and M.A.B.E were in part supported by MARS-Magee Womens Auxiliary. Authors would like to thank Rob J. Munroe and Christian Abratte for help with the production of Grin2A knockout mice. Empire State Stem Cell Fund Contract Number C029155 in part supported this effort. The funders had no role in study design, data collection and analysis, decision to publish, or preparation of the manuscript.

## Author contributions

Conceptualization, T.M.R., J.W.N., K.E.B., M.A.B.-E., and S.L.; formal analysis, T.M.R., J.W.N., S.L.; investigation, T.M.R., J.W.N., A.B.F., K.E.B., R.Y.D, S.S., V.R., B.N., S.L.; writing— original draft, T.M.R., K.E.B., S.L.; writing—review and editing: T.M.R., J.W.N., R.Y.D., M.A.B.-E., S.L.; supervision: M.A.B.-E. and S.L.; funding acquisition: S.L.

## Competing interests

The authors declare no competing interests.
