## [Peer Review File · Nature Communications]

Reviewer #2:

Remarks to the Author:

Related to my earlier comment that the manuscript would strengthen if more specificity to brain sites was provided: the authors should present and discuss the fact the SIRT1 KO mice are much smaller than wild type mice (see discussion: Science. 2006 Mar 17;311(5767):1553-4).

Thank you for raising this point. We have added text discussing the smaller size of the SIRT1 BSKO mice on page 7, as well as a reference on the impact of body weight on susceptibility to ABA¹.

Body weight and fat mass should be reported. Moreover SIRT1 overexpression is known to protect against diet-induced obesity. This complicates the comparison because body composition has a major impact on susceptibility to develop activity-based anorexia. Moreover SIRT1 is expressed ubiquitously and thus besides the proposed mechanism, there could be other mechanisms explaining the effects of systemic manipulations.

We understand the impact of body weight and fat mass in the different models of alimentary disorders, including AN. In Supp. Fig. 2b we present the baseline body weight in each of the phenotypes. Unfortunately, we did not perform the analysis of the fat mass in these animals. In the revised manuscript we include a section of page 7 explaining that body weight differences have a major impact on susceptibility to ABA¹. We also discussed the role of SIRT1 in modulating weight in non-ABA experiments on pages 14 and 15 and the need to run more experiments to determine the exact function of SIRT1 during in the ABA model.

There are contrasting data related to overexpressing or deleting SIRT in brain. For instance deletion of SIRT1 in AgRP neurons results in a lean body mass (Dietrich et al J Neurosci 2010) and overexpression in arcuate nucleus neurons (POMC and AgRP) (Sasaki et al Diabetologia. 2014 Apr;57(4):819-31) protects against age related weight gain (more in line with this study). Besides this numerous other tissues could explain the effect of manipulating SIRT1 at systemic level. Nevertheless, it remains intriguing that SIRT1 inhibition could be a treatment for AN.

We agree with the reviewer that overexpression or knockout in different cells could induce different phenotypes. To minimize confusion, we added a paragraph in the discussion of page 14 to inform the reader of SIRT1's role in endocrine signaling via the HPA axis²⁻⁵. We also discussed the modulation of SIRT1 and its effect on weight as well as the need for future experiments on SIRT1's role in modulating ABA phenotypes.

The authors do not provide evidence that it is only because SIRT1 acts via Grin2A that anorectic behavior is affected.

In the new version of the manuscript on page 14, we discuss other potential mechanisms by which SIRT1 could affect anorectic behaviors.

Minor comments:

1. Spelling should be checked throughout: should Neruo-2A cells be Neuro-2A? page 5 another study found and increase.... etc

We have fixed the noted mistakes as well as others we had previously missed.

2. in the summary AN characteristics are described. It would be better to stay closer to the specific criteria of AN in DSM5.

We agree that our introduction to AN should be more concise and have updated the summary to reflect the DSM5 definition.

3. Summary:animals protects from AN. animals do not have AN, at best they show some features (anorexia, hyperactivity, body weight loss)

We also think that the previous language was misleading and have updated the summary to reflect that the animals were protected from the phenotypes of the ABA model.

4. Main text: what is LGBTQ? I think the general denominator is that sports that encourage lean shape and body mass not judge-based sports

We have changed the text to reflect the pressure on women in activities and sports that encourage lean shape and body weight rather than judge-based sports and removed LGBTQ.

5. page 9 ref 42 is a case report, thus syndromes should be syndrome

Thank you for pointing out our mistake, we have corrected the text.

6. in the discussion the authors mention that AN patients become addicted to hyperactivity and anxiety. No reference is provided. I know AN patients can have an uncontrolled drive for hyperactivity which they cannot stop but that is not an addiction. An addiction to anxiety sounds odd, that cannot be true

We edited the paragraph and relaxed the claims. We have changed the text to explain that the uncontrolled drive parallels features of addiction and have added a reference⁶.

7. p 15 last line: I think potent should be potential

Thank you for pointing out our mistake, it has been corrected.

Reviewer #4:

Remarks to the Author:

The manuscript by Robinette et al. suggests that “SIRT1 accelerates the progression of anorexia nervosa”. However, there are no anorexia nervosa subjects in the studies, which are actually mouse studies. The authors show that SIRT1 inhibition (genetically or pharmacologically) reduces ABA, while overexpression of SIRT1 increases the ABA phenotype. They also present data suggesting that SIRT1 interacts with the transcription factor NRF1 to reduce expression of Grin2A. This work is very interesting and provides novel mechanisms that regulate ABA, and may have implications for anorexia nervosa.

We would like to thank the reviewer for taking time to analyze and comment on our manuscript. We hope that the modifications we have made to improve the manuscript alleviate the concerns surrounding the paper.

1) This reviewer strongly suggests not substituting the word “ABA” with “anorexia nervosa” in the manuscript. ABA is a very good model which may help provide insight into the mechanisms of anorexia nervosa, but the title is highly misleading, and the language in the manuscript makes the leap from ABA to anorexia too often.

We agree that the title and text were misleading and have changed the title and ensured that we no longer jump from AN to ABA when discussing our experiments.

2) The results from all of the measures obtained in an ABA study are not shown for any study in the paper. For example, Figure 1 only shows bodyweight loss, when there were also data for survival, running, and food intake. All measures should be considered to draw conclusions from an ABA study.

In Figure 2, we now report all the data that we have from the ABA study on SIRT1 modulated mice.

We optimized our ABA experiments for the WT mice to lose about 85% of their body weight over the 7 days of the experiment to prevent mice from dying or needing to be pulled from the experiment. On

occasion, a mouse would either die or drop below 30% of their original bodyweight and would be removed. However, it was not enough to warrant a survival curve. We have added to the methods on page 16 to better inform the readers about our experimental setup.

3) On page 5, both male and female mice, young and old, can develop ABA.

Thank you for pointing this out, we have corrected our mistake.

4) The authors state that SIRT1OX mice show an exaggerate increase in physical activity in Figure 2f, but any effect is basely apparent.

We have edited the sentence.

5) ANOVAs are often not used to analyze ABA data, because animals drop out of the study when they lose a predetermined amount of weight (for example, 20% of their initial bodyweight). The high number of missing values as the experiment progresses normally precludes the use of ANOVA. How did the authors avoid this problem?

As mentioned in our response to comment 2, we optimized our ABA experiments for the WT mice to only lose about 85% of their bodyweight. According to the Methods in Molecular Biology Protocol for the Activity-Based Anorexia Mouse Model⁷, we used the two-way ANOVA type-III sums of squares for our unbalanced data. We also state the specific test in our methods on page 20.

6) Figure 2d does not support the statement that SIRT1KO mice are less active.

Thank you for raising this point. In this case the animals had a sharp decrease in activity a few days into the experiment due to physical exhaustion, wasting, and other pathologies associated with ABA⁷. This has been added to the text on page 7.

7) The effects of G2A KO on ABA are quite modest.

We agree that the effects of G2A KO are modest and, in the discussion, suggest two other pathways for future study to fully explain the effects we see from SIRT1 on ABA.

8) Despite some nice findings that SIRT1 modulates ABA, the manuscript does not strongly implicate SIRT1 in anorexia nervosa. This suggestion should be softened. There are likely numerous genes that regulate ABA in rodents, but not all of them will turn out to be disease genes for anorexia in humans. Small scale genetics studies have not replicated well, so we need to wait for GWAS, or large-scale studies looking at rare variants, to find the genes that cause anorexia in humans.

The claims have been softened and we have tempered our language surrounding SIRT1, ABA, and AN throughout the manuscript. Additionally, we have added text in the discussion on page 15 to ensure that readers are aware that future studies will need to be completed to implicate SIRT1 in AN in humans.

- 1 Pjetri, E. *et al.* Identifying predictors of activity based anorexia susceptibility in diverse genetic rodent populations. *PLoS One* **7**, e50453, doi:10.1371/journal.pone.0050453 PONE-D-12-23893 [pii] (2012).
- 2 Boutant, M. & Canto, C. SIRT1 metabolic actions: Integrating recent advances from mouse models. *Mol Metab* **3**, 5-18, doi:10.1016/j.molmet.2013.10.006 S2212-8778(13)00104-X [pii] (2014).
- 3 Yamamoto, M. & Takahashi, Y. The Essential Role of SIRT1 in Hypothalamic-Pituitary Axis. *Front Endocrinol (Lausanne)* **9**, 605, doi:10.3389/fendo.2018.00605 (2018).
- 4 Dietrich, M. O. *et al.* *Agrp* neurons mediate Sirt1's action on the melanocortin system and energy balance: roles for Sirt1 in neuronal firing and synaptic plasticity. *J Neurosci* **30**, 11815-11825, doi:10.1523/JNEUROSCI.2234-10.2010 30/35/11815 [pii] (2010).
- 5 Sasaki, T. *et al.* Hypothalamic SIRT1 prevents age-associated weight gain by improving leptin sensitivity in mice. *Diabetologia* **57**, 819-831, doi:10.1007/s00125-013-3140-5 (2014).
- 6 Barbarich-Marsteller, N. C., Foltin, R. W. & Walsh, B. T. Does anorexia nervosa resemble an addiction? *Curr Drug Abuse Rev* **4**, 197-200, doi:BSP/CDAR/E-Pub/000046 [pii] (2011).
- 7 Klenotich, S. J. & Dulawa, S. C. The activity-based anorexia mouse model. *Methods Mol Biol* **829**, 377-393, doi:10.1007/978-1-61779-458-2_25 (2012).

Reviewers' comments:

Reviewer #2 (Remarks to the Author):

The authors provide a set of data that support that inhibition of SIRT1 is protective in developing anorectic behavior in the ABA model. They went on in exploring the mechanism of action and propose that SIRT1 affects Grin2A and that via Grin2A hyperactivity is affected. It remains unclear what precise bioinformatic analysis resulted in identification of Grin2A (details are not provided), but the set of experiments support a role of Grin2A.

Reviewer #4 (Remarks to the Author):

The authors have addressed all of my comments very well. This is excellent work.

Reviewer #5 (Remarks to the Author):

Since my speciality is in genetics and GWAS methods, I would like to comment specifically on this section of the manuscript.

The authors describe a discovery and replication analysis- the discovery dataset includes 1001 patients from the wang et al 2011 GWAS. They then replicate in an Italian cohort (I assume one of the WTCCC cohorts from the Boraska et al GWAS) using Taqman genotyping of two variants. I have a few questions that I feel would strengthen and clarify this section of the manuscript: The methods section does not provide enough detail to assess the analysis carried out, especially with regards to the discovery analysis. I would like to see descriptions of:

1. The discovery cohort. The discovery sample includes 1001 cases (from dbgap) and 1987 controls. I am unclear as to where the controls come from- are they part of the same study? The number is smaller than reported in the original Wang et al GWAS. It would be useful to see more details about these cohorts- I could not find any information in the methods section. For example, are these individuals broadly matched to 'european' descent? Or are they matched to specific country of origin?

2. Correction for covariates. Even if samples are matched for ancestry, population stratification may still occur. I note that 'statistics was corrected for age, sex and race'. What does 'race' mean, in this context? Self-defined? Genotype-derived principal components? It would also be helpful to know the distributions of all of these covariates among cases and controls.

3. Genotyping chips used. Are cases and controls from the discovery cohort genotyped on the same chip? If not, what QC was done to ensure that there were no differences induced due to differing chip chemistry between the two datasets?

4. SNP selection. I see that 7 SNPs were selected from the discovery cohort for analysis. How were these selected? Proximity to SIRT1? How were the boundaries of the region defined? Are there other SNPs in this region that were not included? Were these SNPs directly genotyped, or imputed?

5. QC. The authors mention using plink and R to create a linear regression model. Was there any QC using plink, prior to this, to correct for call rate, missingness, genotyping quality or imputation quality?

Results:

The authors present two stages of analysis here: first, a description of association statistics among the 7 variants; second, a replication analysis in a second cohort.

This analysis is treated as a 'candidate gene' type analysis, testing and correcting for only a small

number of SNPs. I do not think this analysis is without merit- there seems strong biological arguments for investigating this gene, and I am encouraged by the replication (and consistent direction of effect) across two studies. However, it is very difficult to assess the analysis carried out given the sparsity of the description of the methods (below, I include a list of items that I would like to see included in the methods section).

Regarding the analytical approach:

Most importantly:

I looked at the original Wang et al GWAS to compare association levels there, to the data presented here. It appears that the SNPs in this locus do not reach the same level of significance in the original study as they do in this analysis. (Eg, in figure 1 in Wang et al, the manhattan plot for chr10 is clearly far below the p-values reported here). This is very concerning. Could the authors comment on why this may be?

Some other thoughts on the analytical approach:

1. Why was the CHOP GWAS from 2011 chosen, rather than any of the three more recent, and larger GWAS (Boraska, Huckins, Duncan?). These studies have larger sample sizes, and presumably use more recent imputation methods. These should provide more SNPs to test and greater power. The largest, PGC GWAS (Duncan et al) summary statistics are freely available (<https://www.med.unc.edu/pgc/results-and-downloads/>). It would be edifying to look up these SNPs (or, ideally, the full locus), in this PGC GWAS.
2. It is unusual to select SNPs from a genome-wide study and to present these in a candidate gene-style analysis. Technically, I do believe they should be subjected to genome-wide significance thresholds ($5e-08$). After all, the data are not 'unexamined'- the authors are not truly agnostic to the genome-wide association study results. If the gene had been significant in the GWAS, I cannot believe they would not simply have cited it, rather than re-analysing this subset of SNPs. However, the fact that these SNPs survive permutation analysis, and replicate in a second cohort is good. I think replication is crucial given the lowered p-value threshold, so I would not present the second SNP (rs109..) as significant. I also find the description of the LD block strange- it is not unusual to find GWAS loci within high LD blocks, and I don't think this adds much to the text (especially given non-replication).
3. A more convincing analysis would include a comparison of these SNPs across all AN GWAS (I realise there is substantial overlap in cases from these studies, which complicates this), perhaps using a forest plot or similar. I would be more convinced of the relevance of these variants were I to see replication across these studies, and consistent effect sizes, etc. (Incidentally, it would be nice to see odds ratios/betas in Fig1C).
4. Given that the interest in this study is a particular gene, why not perform some kind of burden testing? ie, an aggregation of these variants to achieve an SIRT1 association statistic?

Reviewer 2

The authors provide a set of data that support that inhibition of SIRT1 is protective in developing anorectic behavior in the ABA model. They went on in exploring the mechanism of action and propose that SIRT1 affects Grin2A and that via Grin2A hyperactivity is affected. It remains unclear what precise bioinformatic analysis resulted in identification of Grin2A (details are not provided), but the set of experiments support a role of Grin2A.

Reviewer 4

The authors have addressed all of my comments very well. This is excellent work.

Reviewer 5

Since my specialty is in genetics and GWAS methods, I would like to comment specifically on this section of the manuscript.

The authors describe a discovery and replication analysis- the discovery dataset includes 1001 patients from the Wang et al 2011 GWAS. They then replicate in an Italian cohort (I assume one of the WTCCC cohorts from the Boraska et al GWAS) using Taqman genotyping of two variants. I have a few questions that I feel would strengthen and clarify this section of the manuscript:

The methods section does not provide enough detail to assess the analysis carried out, especially with regards to the discovery analysis. I would like to see descriptions of:

1. The discovery cohort. The discovery sample includes 1001 cases (from dbgap) and 1987 controls. I am unclear as to where the controls come from- are they part of the same study? The number is smaller than reported in the original Wang et al GWAS. It would be useful to see more details about these cohorts- I could not find any information in the methods section. For example, are these individuals broadly matched to 'European' descent? Or are they matched to specific country of origin?

The 1987 controls came from a GWAS study from Landi et. al.¹. They were chosen based on a dbGaP search for other GWASs that used the Illumina HumanHap610 platform. For our replication experiment, we genotyped the SNPs ourselves via the TaqMan method as opposed to using a chip. For the 3978 controls, we pooled data from Firmann et. al.² and Simon-Sanchez et. al.³. We have added the explanations into our text on page 3 and method section on page 20.

2. Correction for covariates. Even if samples are matched for ancestry, population stratification may still occur. I note that 'statistics was corrected for age, sex and race'. What does 'race' mean, in this context? Self-defined? Genotype-derived principal components? It would also be helpful to know the distributions of all of these covariates among cases and controls.

We now reference each of the studies in the methods regarding how each study determined the sample's ancestry of origin¹⁻⁵ on page 20. E.g. for the 1001 cases, PLINK software was used for MDS on markers not in linkage disequilibrium to identify ancestry. Individuals that were genetically inferred to not be European were removed⁴.

3. Genotyping chips used. Are cases and controls from the discovery cohort genotyped on the same chip? If not, what QC was done to ensure that there were no differences induced due to differing chip chemistry between the two datasets?

For the 1001 cases and 1987 controls, the Illumina HumanHap610 platform was used^{1,4}. For the 112 cases in the Italian study, the TaqMan method was used. The 3937 controls were genotyped with either the Affimetrix 500 K SNP chip², HumanHap550 v1 beadchips, HumanHap550 v3 beadchips, or a combination of HumanHap300 and HumanHap240S beadchips³. We have added these details to our methods section on page 17.

4. SNP selection. I see that 7 SNPs were selected from the discovery cohort for analysis. How were these selected? Proximity to SIRT1? How were the boundaries of the region defined? Are there other SNPs in this region that were not included? Were these SNPs directly genotyped, or imputed?

We now explain in the text page 20 that we selected the SIRT1 SNPs due to their prior association with metabolic disorders⁶⁻¹². The SIRT1 SNPs were directly genotyped; those that were associated with metabolic disorders, but not on the chip were excluded.

5. QC. The authors mention using plink and R to create a linear regression model. Was there any QC using plink, prior to this, to correct for call rate, missingness, genotyping quality or imputation quality?

Yes, we required Hardy-Weinberg p values $> 10^{-7}$, SNP missingness $< 30\%$, individual missingness $< 10\%$, and minor allele frequency > 0.01 and have added this analysis in our methods on page 20.

Results:

The authors present two stages of analysis here: first, a description of association statistics among the 7 variants; second, a replication analysis in a second cohort.

This analysis is treated as a 'candidate gene' type analysis, testing and correcting for only a small number of SNPs. I do not think this analysis is without merit- there seems strong biological arguments for investigating this gene, and I am encouraged by the replication (and consistent direction of effect) across two studies. However, it is very difficult to assess the analysis carried out given the sparsity of the description of the methods (below, I include a list of items that I would like to see included in the methods section).

Regarding the analytical approach:

Most importantly:

6. I looked at the original Wang et al GWAS to compare association levels there, to the data presented here. It appears that the SNPs in this locus do not reach the same level of significance in the original study as they do in this analysis. (Eg, in figure 1 in Wang et al, the manhattan plot for chr10 is clearly far below the p-values reported here). This is very concerning. Could the authors comment on why this may be?

Due to Wang et al.'s underpowered dataset⁴, no significant SNPs were found in the study. However, because both ours and others work implicated SIRT1 in psychiatric and metabolic disorders, we pursued SIRT1 as a candidate gene. When completing the SNP analysis on the seven SIRT1 SNPs, we found that two of the SNPs were significantly associated with AN. Upon replication only one of the SNPs remained significant. Requiring candidate gene studies to meet the statistical criteria for genome-wide significance is considered overly stringent¹³, especially since Wang et. al. had not found any gene effects⁴.

Some other thoughts on the analytical approach:

7. Why was the CHOP GWAS from 2011 chosen, rather than any of the three more recent, and larger GWAS (Boraska, Huckins, Duncan?). These studies have larger sample sizes, and presumably use more recent imputation methods. These should provide more SNPs to test and greater power. The largest, PGC GWAS (Duncan et al) summary statistics are freely available (<https://www.med.unc.edu/pgc/results-and-downloads/>). It would be edifying to look up these SNPs (or, ideally, the full locus), in this PGC GWAS.

At the time we completed the analysis, the CHOP GWAS was the only study available. However, based on the comments of a previous reviewer, we had submitted an analysis proposal for access to the PGC GWAS data¹⁴ with a member of the PGC and the co-chair of their Eating Disorders Workgroup. Unfortunately, because they were preparing for their own primary publication¹⁵, they were not accepting any new proposals. However, after discussion with their co-chair, we decided it was fine to move forward with our data because the majority of our 1113 individuals overlapped with their 3495 individuals.

8. It is unusual to select SNPs from a genome-wide study and to present these in a candidate gene-style analysis. Technically, I do believe they should be subjected to genome-wide significance thresholds (5×10^{-8}). After all, the data are not 'unexamined'- the authors are not truly agnostic to the genome-wide association study results. If the gene had been significant in the GWAS, I cannot believe they would not simply have cited it, rather than re-analysing this subset of SNPs. However, the fact that these SNPs survive permutation analysis, and replicate in a second cohort is good. I think replication is crucial given the lowered p-value threshold, so I would not present the second SNP (rs109..) as significant. I also find the description of the LD block strange- it is not unusual to find GWAS loci within high LD blocks, and I don't think this adds much to the text (especially given non-replication).

We edited the paragraph on page 3 and specified that only rs730821 was significantly associated with AN in the replication study. Also, we had decided to show the LD plot for other groups to compare to if our study is replicated. We now indicate this in the text on page 3 when discussing the lack of replication.

9. A more convincing analysis would include a comparison of these SNPs across all AN GWAS (I realize there is substantial overlap in cases from these studies, which complicates this), perhaps using a forest plot or similar. I would be more convinced of the relevance of these variants were I to see replication across these studies, and consistent effect sizes, etc. (Incidentally, it would be nice to see odds ratios/betas in Fig1C).

Due to the underpowered nature of the previous GWAS studies, as mentioned in comment 6 above, a comparison of these studies would not yield any appreciable results. We have added the odds ratios to Figure 1C on page 4.

10. Given that the interest in this study is a particular gene, why not perform some kind of burden testing? ie, an aggregation of these variants to achieve an SIRT1 association statistic?

The MAF of the SIRT1 variants were >1%, as stated in our selection criteria in comment 5, and since burden tests are typically completed on rare variants (<1% MAF), we decided to continue with the candidate gene analysis.

References

- 1 Landi, M. T. *et al.* A genome-wide association study of lung cancer identifies a region of chromosome 5p15 associated with risk for adenocarcinoma. *Am J Hum Genet* **85**, 679-691, doi:10.1016/j.ajhg.2009.09.012 (2009).
- 2 Firmann, M. *et al.* The CoLaus study: a population-based study to investigate the epidemiology and genetic determinants of cardiovascular risk factors and metabolic syndrome. *BMC Cardiovasc Disord* **8**, 6, doi:10.1186/1471-2261-8-6 (2008).
- 3 Simon-Sanchez, J. *et al.* Genome-wide association study reveals genetic risk underlying Parkinson's disease. *Nat Genet* **41**, 1308-1312, doi:10.1038/ng.487 (2009).
- 4 Wang, K. *et al.* A genome-wide association study on common SNPs and rare CNVs in anorexia nervosa. *Mol Psychiatry* **16**, 949-959, doi:10.1038/mp.2010.107mp2010107 [pii] (2011).
- 5 Cellini, E. *et al.* Glucocorticoid receptor gene polymorphisms in Italian patients with eating disorders and obesity. *Psychiatr Genet* **20**, 282-288, doi:10.1097/YPG.0b013e32833a2142 (2010).
- 6 Knoll, N. *et al.* Gene set of nuclear-encoded mitochondrial regulators is enriched for common inherited variation in obesity. *PLoS One* **8**, e55884, doi:10.1371/journal.pone.0055884 (2013).
- 7 Zillikens, M. C. *et al.* SIRT1 genetic variation is related to BMI and risk of obesity. *Diabetes* **58**, 2828-2834, doi:10.2337/db09-0536 (2009).
- 8 Clark, S. J. *et al.* Association of sirtuin 1 (SIRT1) gene SNPs and transcript expression levels with severe obesity. *Obesity (Silver Spring)* **20**, 178-185, doi:10.1038/oby.2011.200oby2011200 [pii] (2012).
- 9 Rai, E. *et al.* The interactive effect of SIRT1 promoter region polymorphism on type 2 diabetes susceptibility in the North Indian population. *PLoS One* **7**, e48621, doi:10.1371/journal.pone.0048621 (2012).
- 10 Maeda, S. *et al.* Association between single nucleotide polymorphisms within genes encoding sirtuin families and diabetic nephropathy in Japanese subjects with type 2 diabetes. *Clin Exp Nephrol* **15**, 381-390, doi:10.1007/s10157-011-0418-0 (2011).
- 11 Zhao, Y. *et al.* SIRT1 rs10823108 and FOXO1 rs17446614 responsible for genetic susceptibility to diabetic nephropathy. *Sci Rep* **7**, 10285, doi:10.1038/s41598-017-10612-7 (2017).
- 12 Shimoyama, Y., Suzuki, K., Hamajima, N. & Niwa, T. Sirtuin 1 gene polymorphisms are associated with body fat and blood pressure in Japanese. *Transl Res* **157**, 339-347, doi:10.1016/j.trsl.2011.02.004 (2011).
- 13 Patnala, R., Clements, J. & Batra, J. Candidate gene association studies: a comprehensive guide to useful in silico tools. *BMC Genet* **14**, 39, doi:10.1186/1471-2156-14-39 (2013).
- 14 Duncan, L. *et al.* Significant Locus and Metabolic Genetic Correlations Revealed in Genome-Wide Association Study of Anorexia Nervosa. *Am J Psychiatry* **174**, 850-858, doi:10.1176/appi.ajp.2017.16121402 (2017).
- 15 Watson, H. J. *et al.* Genome-wide association study identifies eight risk loci and implicates metabo-psychiatric origins for anorexia nervosa. *Nature Genetics*, doi:10.1038/s41588-019-0439-2 (2019).

REVIEWERS' COMMENTS:

Reviewer #6 (Remarks to the Author):

I think the authors have tried very well to answer the critiques from Reviewer 5 regarding the GWAS study and analysis. Despite this, I think it is clear that because GWAS studies have been underpowered (as the authors note), that these data are not the strongest and that many of the questions raised cannot actually be answered for this reason.

Despite this, I believe that many of these issues would be mitigated if the authors shifted the placement of this data in the manuscript. Specifically, this paper is mainly about the role of SIRT1 in ABA in mice and the authors concede that this is a candidate gene approach. For that reason, it would be preferable to start with the animal work and to end the manuscript showing how SIRT1 relates to the GWAS studies. This way they can temper their claims regarding the significance in the GWAS studies, while still presenting this intriguing data.

On a more detailed note, it is now clear from the rebuttal that the samples chosen (1001 AN vs. 1987 controls and the 112 vs. 3937) come from different studies, which was not clear beforehand. This does not make sense to me -- there are control groups in each of the eating disorders studies. What is the rationale for using another control group instead of the original? And how was this unrelated control group chosen?

REVIEWERS' COMMENTS:

Reviewer #6 (Remarks to the Author):

I think the authors have tried very well to answer the critiques from Reviewer 5 regarding the GWAS study and analysis. Despite this, I think it is clear that because GWAS studies have been underpowered (as the authors note), that these data are not the strongest and that many of the questions raised cannot actually be answered for this reason.

Despite this, I believe that many of these issues would be mitigated if the authors shifted the placement of this data in the manuscript. Specifically, this paper is mainly about the role of SIRT1 in ABA in mice and the authors concede that this is a candidate gene approach. For that reason, it would be preferable to start with the animal work and to end the manuscript showing how SIRT1 relates to the GWAS studies. This way they can temper their claims regarding the significance in the GWAS studies, while still presenting this intriguing data.

We agree that this suggested order of results is preferable to prevent the results of our paper from being overinterpreted. Our candidate gene analysis is now in the supplemental information and discussed at the end of the paper.

On a more detailed note, it is now clear from the rebuttal that the samples chosen (1001 AN vs. 1987 controls and the 112 vs. 3937) come from different studies, which was not clear beforehand. This does not make sense to me -- there are control groups in each of the eating disorders studies. What is the rationale for using another control group instead of the original? And how was this unrelated control group chosen?

The 1001 AN GWAS (Wang et. al. 2010) was from a pool of subjects from multiple sources with a maximum age of 45, and an average of 27 years, with the average age of first symptom of 15.1 years. However, the controls from the Wang. et. al. study were pediatric, with an average age of 12.75 years. This causes two main issues, one being that the average control age is below the age of onset, so we could not be sure that the controls would not develop AN at a later time. Additionally, it prevents us from being able to age-match the samples.

For our verification cohort of 112 AN cases, we were only able to obtain the case samples from the Nacmias lab and we genotyped them for rs730821 and rs10997881 in our lab. We had to use an outside control group due to the control samples not being available to us.